# In Vitro Synergism of Penicillin and Ceftriaxone against *Enterococcus faecalis*

**DOI:** 10.3390/microorganisms9102150

**Published:** 2021-10-14

**Authors:** Lara Thieme, Simon Briggs, Eamon Duffy, Oliwia Makarewicz, Mathias W. Pletz

**Affiliations:** 1Institute for Infectious Diseases and Infection Control, Jena University Hospital, Friedrich Schiller University, 07747 Jena, Germany; oliwia.makarewicz@med.uni-jena.de (O.M.); mathias.pletz@med.uni-jena.de (M.W.P.); 2Leibniz Center for Photonics in Infection Research, Jena University Hospital, Friedrich Schiller University, 07747 Jena, Germany; 3Infectious Disease Unit, Auckland City Hospital, Auckland 1023, New Zealand; SBriggs@adhb.govt.nz (S.B.); EamonD@adhb.govt.nz (E.D.)

**Keywords:** *E. faecalis* endocarditis, synergism, OPAT, penicillin/ceftriaxone, ampicillin/ceftriaxone, checkerboard

## Abstract

*Enterococcus faecalis* infective endocarditis is commonly treated with intravenous ampicillin/ceftriaxone combination therapy. Ampicillin, however, is unsuitable for outpatient parenteral antibiotic therapy (OPAT) regimens due to its instability in 24 h continuous infusors, and has been successfully replaced by benzylpenicillin used together with ceftriaxone in a few small case series. Since in vitro synergy data of penicillin/ceftriaxone against *E. faecalis* are still lacking, checkerboard assays were performed for 28 clinical *E. faecalis* isolates and one laboratory standard strain. Synergistic effects (both lowest and median FICI) were observed for penicillin/ceftriaxone in 15/29 isolates, while ampicillin/ceftriaxone exhibited synergism in 22/29 isolates. For isolates with ceftriaxone MICs ≤ 256 mg/L, the addition of free ceftriaxone trough concentrations to penicillin or ampicillin resulted in comparable synergistic effects for both combinations. In contrast, for isolates with ceftriaxone MICs ≥ 512 mg/L free ceftriaxone trough concentrations were only sufficient to exhibit synergistic effects in combination with ampicillin, but not penicillin. This study suggests that benzylpenicillin/ceftriaxone would be expected to be suitable for the OPAT treatment of enterococcal endocarditis for *E. faecalis* isolates with ceftriaxone MICs ≤ 256 mg/L. However, combination therapy would be expected to provide no advantage over benzylpenicillin monotherapy for isolates with ceftriaxone MICs ≥ 512 mg/L. Further investigation is required to analyse the relationship between ceftriaxone susceptibility and penicillin/ceftriaxone synergy, especially for isolates with ceftriaxone MICs of 64 to 512 mg/L.

## 1. Introduction

*Enterococcus faecalis* is an increasingly common cause of infective endocarditis (IE) and should be treated with prolonged synergistic, bactericidal antibiotic combination therapy [1,2]. Contemporary treatment guidelines recommend the use of intravenous (IV) ampicillin, amoxicillin, or benzylpenicillin combined with IV gentamicin for gentamicin susceptible isolates, or IV ampicillin combined with IV ceftriaxone for both high-level aminoglycoside-resistant (HLAR) and non-HLAR isolates [2,3]. Ceftriaxone is a more attractive synergy antibiotic than gentamicin, since it causes significantly fewer adverse effects [1,4].

Although a previous guideline for the Outpatient Parenteral Antibiotic Therapy (OPAT) treatment of IE was restrictive [5], a recent prospective cohort study has shown that the OPAT treatment of a broader range of patients with IE provided excellent results [6]. The abovementioned guideline-recommended regimens for the treatment of *E. faecalis* IE (EFIE) may be challenging to administer via an OPAT service due to the multiple doses required per day. The administration of the treatment by community nurses is often limited to once or twice daily injections or changing a 24 h elastomeric continuous infusor. While ampicillin/ceftriaxone has been proposed as an OPAT regimen for EFIE utilising either elastomeric continuous infusors or programmable pumps for the delivery of ampicillin [7], the literature shows conflicting results regarding the stability of ampicillin in elastomeric continuous infusors [8,9,10,11]. While ampicillin, which cannot be applied orally for reasons of bioavailability, could theoretically be replaced by orally administered amoxicillin, the clinical evidence supporting oral antibiotic regimens for treating EFIE is scarce. The use of oral antibiotics as part of the treatment of endocarditis has only been assessed in one trial [12] that enrolled a tightly defined group of patients with left-sided endocarditis who were transitioned to oral regimens. The patients with EFIE were treated with a variety of oral antibiotic combinations, the most common of which were amoxicillin /moxifloxacin, amoxicillin/linezolid, and amoxicillin/rifampicin. Oral amoxicillin/IV ceftriaxone has not been tested in a trial situation, as far as we are aware, and should therefore only be administered as part of a very carefully designed trial.

IV benzylpenicillin has been used as an alternative to IV ampicillin/amoxicillin for the treatment of EFIE, since benzylpenicillin is more stable in elastomeric continuous infusors and therefore better suited for OPAT regimens. The use of intravenous benzylpenicillin/ceftriaxone for the OPAT-guided treatment of *E. faecalis* endocarditis is common in New Zealand and also occurs in Australia and America. Benzylpenicillin has been successfully combined with ceftriaxone for the OPAT treatment of enterococcal endocarditis in a few small clinical case series [13,14,15], but in vitro synergy data supporting the use of benzylpenicillin with ceftriaxone for OPAT regimens are still lacking.

Here, we report the checkerboard synergy analysis of penicillin/ceftriaxone and ampicillin/ceftriaxone in 28 clinical *E. faecalis* isolates and one laboratory standard strain, allowing for a comparison of both combinations at concentrations achieved by OPAT dosage regimens.

## 2. Materials and Methods

### 2.1. Enterococcal Strains and Antibiotics

Clinical *E. faecalis* isolates were obtained from blood cultures from, or the heart valves of, patients with enterococcal infection at various foci, including eight patients with infective endocarditis (Table 1). The antibiotic-susceptibility data for all clinical isolates are depicted in Appendix A. All patients were admitted to Jena University Hospital, Germany, in 2015. *E. faecalis* ATCC 29,212 served as a reference strain. Bacterial liquid cultures were prepared in Todd Hewitt broth (Karl Roth, Karlsruhe, Germany). Test solutions of ampicillin (AMP) (Karl Roth, Karlsruhe, Germany), ceftriaxone (CRO) (TCI Europe, Zwijndrecht, Belgium), and benzylpenicillin (PEN) (InfectoPharm, Heppenheim, Germany) were prepared immediately before use.

### 2.2. Synergism Testing by Checkerboard Assays

Checkerboard assays for penicillin/ceftriaxone were performed as described previously [17], with 11 and 7 serial 2-fold dilution steps for benzylpenicillin and ceftriaxone, respectively. The benzylpenicillin concentrations tested were chosen to cover the penicillin breakpoint range. The ceftriaxone concentrations tested included those that approximate the range of estimated or measured mean free plasma ceftriaxone trough concentrations expected to be achieved with OPAT ceftriaxone-dosing regimens of 2 g every 12 h, 4 g once daily, or 2 g once daily (4, 1.5, and 1 mg/L, respectively) [18,19,20], hereafter referred to as “free ceftriaxone trough concentrations”. The checkerboard assays used for ampicillin/ceftriaxone were partly assessed in a previous study [21]; however, they were newly evaluated according to novel EUCAST guidelines [16] (see Table 1, footnote c), which has led to a change in MICs and fractional inhibitory concentrations indices (FICI) for 10 out of 21 previously tested isolates. Checkerboards for ampicillin/ceftriaxone were repeated for eight of these isolates to guarantee the reproducibility of previous results. Each checkerboard assay was performed in duplicate. The effects of the combined antibiotics were evaluated by calculating the FICI along the turbidity/non-turbidity interface using the following formula:$${\mathrm{FICI}}_{\mathrm{penicillin}/\mathrm{ceftriaxone}}=\frac{\mathrm{MIC}\;\mathrm{penicillin}\;\left(\mathrm{combination}\right)}{\mathrm{MIC}\;\mathrm{penicillin}\;\left(\mathrm{alone}\right)}+\frac{\mathrm{MIC}\;\mathrm{ceftriaxone}\;\left(\mathrm{combination}\right)}{\mathrm{MIC}\;\mathrm{ceftriaxone}\;\left(\mathrm{alone}\right)}.$$

Three different methods for interpreting the FICIs were used: (i) the lowest FICI value, with values ≤ 0.5 indicating synergism and 0.5 < FICI < 4 indicating no interaction; (ii) the median FICI value, taking 0.8 as the synergy threshold; (iii) single, one-point FICIs at free ceftriaxone trough concentrations.

### 2.3. Statistical Analysis

The correlation of the MICs and FICIs was tested using the nonparametric Spearman’s rank-correlation coefficient (r_s_) with a two-tailed CI of 95%.

## 3. Results

Synergistic effects were observed for penicillin/ceftriaxone in 16 (lowest FICI method) and 21 (median FICI method) of the 29 tested *E. faecalis* strains (Table 1). Ampicillin/ceftriaxone synergised in 22 strains (both lowest and median FICI method) (Table 1). Ampicillin MICs (MIC_AMP_) ranged from 0.25 to 2 mg/L, while penicillin MICs (MIC_PEN_) ranged from 0.5 to 4 mg/L, the highest MIC_PEN_ of 4 mg/L being below the CLSI susceptibility breakpoint of 8 mg/L. Ceftriaxone MICs (MIC_CRO_) ranged from 1–2 to more than 1024 mg/L. The MIC_AMP_ showed no correlation with the MIC_CRO_, but the MIC_PEN_ exhibited a weak positive correlation with the MIC_CRO_ (r_s_ = 0.52, *p* = 0.004). Both the MIC_PEN_ and MIC_CRO_ displayed a strong inverse correlation with the FICI values of penicillin/ceftriaxone (FICI_PEN/CRO_)—i.e., the higher the MIC_PEN_ or MIC_CRO_, the lower the FICI_PEN/CRO_ (MIC_PEN_ r_s_ = −0.61, *p* = 0.001; MIC_CRO_ r_s_ = −0.71, *p* < 0.001). The MIC_CRO_ inversely correlated with the FICI_AMP/CRO_ (r_s_ = −0.76, *p* < 0.001), but the MIC_AMP_ did not.

To analyse whether the synergistic effects determined by the lowest and median FICI methods were achieved at free ceftriaxone trough concentrations, the lowest sub-MIC ceftriaxone concentrations that resulted in a reduction in the MIC_PEN_ or MIC_AMP_ to one half, one quarter, and one eighth of the MIC_PEN_ or MIC_AMP_ are recorded in Table 2, together with the corresponding single, one-point FICIs. For 13 isolates (9367; 1653; 848; 5597; 6886; 67,230; 2164; 4497; 11,223; 22,424; 10,485; 13,703; 7914), synergy between penicillin/ceftriaxone occurred at free ceftriaxone trough concentrations, while for another 8 isolates (5187; 245; ATCC; 6747; 8669; 281; 26,786; 404), the addition of these free ceftriaxone trough concentrations resulted in a four-fold reduction in the MIC_PEN_ but with FICIs above the synergy threshold of 0.5. In contrast, ampicillin/ceftriaxone exhibited synergism in 23 isolates at free ceftriaxone trough concentrations. For the isolates with MIC_CRO_ ≥ 512 mg/L, lower sub-MIC ceftriaxone concentrations were required in combination with ampicillin than in combination with penicillin to produce the same effect (Table 2). Here, the addition of free ceftriaxone trough concentrations to penicillin resulted in a reduction in the MIC_PEN_ in one of seven isolates, whereas the addition to ampicillin reduced the MIC_AMP_ in six of seven isolates.

## 4. Discussion

The synergy analysis of ampicillin/ceftriaxone and penicillin/ceftriaxone in 29 *E. faecalis* isolates revealed that both combinations have comparable synergistic effects for isolates with MIC_CRO_ ≤ 256 mg/L. A reduction in the MIC_PEN_ with the addition of ceftriaxone concentrations approximating free ceftriaxone trough concentrations was shown in all of these isolates. In contrast, for isolates with MIC_CRO_ ≥ 512 mg/L discordance was seen between the synergy provided by ampicillin/ceftriaxone and by penicillin/ceftriaxone at free ceftriaxone trough concentrations. These ceftriaxone concentrations were insufficient in combination with penicillin, but sufficient in combination with ampicillin to produce a synergistic effect for isolates with an MIC_CRO_ ≥ 512 mg/L.

While a synergistic effect between ampicillin and ceftriaxone in *E. faecalis* is not a new finding, and in fact forms part of the basis for recommending this combination for the treatment of EFIE, there are currently no data available showing a potential synergistic effect between penicillin and ceftriaxone. Importantly, our study shows that a similar synergistic effect between penicillin and ceftriaxone cannot be assumed just because there is a proven synergistic effect between ampicillin and ceftriaxone. Although our sample size is relatively small, with 29 *E. faecalis* isolates, the study on which the current guidelines for the use of ampicillin and ceftriaxone for the treatment of EFIE is based contained just 10 isolates [22]. Our data show that the suitability of penicillin/ceftriaxone for the treatment of EFIE likely depends on the specific isolate’s ceftriaxone susceptibility. However, the interpretation of the relationship between ceftriaxone susceptibility and penicillin/ceftriaxone synergy in this study is limited by the lack of isolates with an MIC_CRO_ of 64 and 128 mg/L, as well as by the limited number of isolates with an MIC_CRO_ of 256 and 512 mg/L. Our *E. faecalis* strain collection comprises 50 clinical isolates collected between 2015 and 2017. The ceftriaxone MIC testing of this collection shows two populations: a larger population of isolates (68%) centred around 8 mg/L ± 1–2× MIC and a smaller population of isolates (32%) centered around 512 mg/L ± 1× MIC (Appendix A). None of our isolates exhibited a ceftriaxone MIC of 64 or 128 mg/L. The EUCAST MIC_CRO_ distribution data for *E. faecalis* show that, of the 8314 submitted clinical isolates, only 5% had an MIC_CRO_ ≤ 32 mg/L [23], which contrasts strongly with the MIC_CRO_ distribution of our cohort.

Caution should be applied when drawing conclusions from microbiological studies for use in the clinical setting. While the checkerboard assay has been intensively used for studying antibiotic interactions, clinical correlation studies linking in vitro synergy data to direct treatment outcomes are lacking. The methodology of the checkerboard assay does have some limitations, such as a high degree of variability in the selection of the wells used for the final FICI calculation [24,25]. To compensate for a potential selection bias leading to an overestimation of the synergistic effect, we used the median FICI, the lowest FICI, and one-point FICIs at free ceftriaxone trough concentrations for a clinically meaningful interpretation of the checkerboard assay.

The synergistic effect of dual beta-lactam therapy is thought to be based on the complementary inhibition of penicillin binding protein (PBP) homologues, resulting in the inhibition of cell-wall synthesis. Little is known about the detailed functions of PBPs in *E. faecalis*, and this understanding is complicated by the inconsistent labelling of the different PBP homologues [26]. The synergism of amoxicillin/cefotaxime in a single *E. faecalis* isolate was postulated to be explained by the partial saturation of the essential PBPs 4 and 5 by amoxicillin, coupled with the complete saturation of the non-essential PBPs 2 and 3 by very low cefotaxime concentrations [27]. Ceftriaxone resistance is known to be mediated by the overproduction and mutations of PBPs 4 and 5 as well as other, novel low-affinity class-B PBPs, further reducing the already low affinity of ceftriaxone for these essential PBPs [26]. The discordance of the synergistic effects between ampicillin/ceftriaxone and penicillin/ceftriaxone in isolates with high MIC_CRO_ might be explained by the more complete saturation of the ceftriaxone-resistance-mediating PBP profile by ampicillin than by penicillin. The difference in saturation would mean that higher ceftriaxone concentrations would be required to compensate for the poorer binding of penicillin to the altered PBPs. This explanation is supported by the fact that the MIC_PEN_, but not the MIC_AMP_, positively correlated with the MIC_CRO_, indicating that both antibiotics target similar PBP homologues. Interestingly, a study with one *E. faecalis* isolate showed no interaction between penicillin/ceftriaxone, but synergistic interaction between penicillin/ceftaroline, which is a novel cephalosporin with enhanced affinity to PBP 5 [28]. This further supports the hypothesis that the incomplete binding of PBP 5 by penicillin and ceftriaxone counteracts the synergistic effect.

The addition of ceftriaxone to benzylpenicillin provides synergy, or at least partial synergy, for *E. faecalis* isolates with an MIC_CRO_ ≤ 256 mg/L. No significant benefit from adding ceftriaxone to benzylpenicillin is expected for any isolate with an MIC_CRO_ ≥ 512 mg/L. These microbiological data support the use of OPAT treatment with the continuous infusion of benzylpenicillin and ceftriaxone for EFIE for isolates with an MIC_CRO_ ≤ 256 mg/L. As the penicillin/ceftriaxone synergy-testing data for isolates with MIC_CRO_ of 64 to 512 mg/L are limited in this study, further investigation is required to establish a reliable MIC_CRO_ cut-off above which the combination of benzylpenicillin and ceftriaxone is not superior to benzylpenicillin alone.

## Figures and Tables

**Table 1 microorganisms-09-02150-t001:** Susceptibility (MIC) and synergy (FICI) results for penicillin/ceftriaxone versus ampicillin/ceftriaxone in the patient cohort.

Isolate ^a^	Clinical Background ^b^	Gender	Age (Years)	MIC CRO (mg/L)	MIC PEN (mg/L)	MIC AMP (mg/L)	FICI ^d^ of PEN/CRO	FICI of AMP/CRO
5187	urosepsis	male	79	1–2 ^c^	1	1	0.75 (1.13)	0.63 (0.88)
245	endocarditis	male	76	2 ^c^	1	0.5	0.75 (1.03)	0.63 (0.88)
26,786	endocarditis	male	67	2	1	0.25	0.56 **(0.75)**	0.74 (1.05)
404	endocarditis	female	78	4	0.5	0.25	0.75 **(0.75)**	**0.49 (0.68)**
ATCC	/	/	/	4	1	1	0.75 **(0.75)**	**0.50 (0.69)**
6747	biliary tract infection	male	81	4 ^c^	1	1	0.75 **(0.75)**	**0.50 (0.63)**
11,223	endocarditis	female	65	8	2	0.5	**0.38 (0.56)**	**0.31 (0.50)**
22,424	endocarditis	male	68	8	1	0.5	**0.25 (0.56)**	**0.37 (0.47)**
8669	OI	male	80	8 ^c^	1	1	0.63 **(0.75)**	**0.25 (0.38)**
9367	recurrent bacteraemia	female	85	8 ^c^	2	2	**0.50 (0.63)**	**0.38 (0.63)**
1653	urosepsis	female	87	8 ^c^	2	1	**0.38 (0.63)**	**0.38 (0.56)**
848	OI	male	60	8	2	1	**0.38 (0.56)**	**0.31 (0.47)**
5597	OI	female	55	8–16 ^c^	2	1	**0.31 (0.52)**	**0.50 (0.63)**
10,485	endocarditis	male	74	16	1	0.5	**0.38** (0.81)	**0.24 (0.53)**
6886	wound infection	female	68	16 ^c^	1	2	0.63 **(0.75)**	**0.31 (0.52)**
281	sepsis	female	79	16 ^c^	1	1	**0.50 (0.63)**	**0.31 (0.38)**
67,230	endocarditis	male	39	16	2	1	**0.38 (0.47)**	**0.25 (0.45)**
2164	OI	female	78	16 ^c^	2	2	**0.25 (0.56)**	**0.38 (0.63)**
4497	urosepsis	female	67	32 ^c^	2	1	**0.31 (0.45)**	**0.38 (0.56)**
10,021	urosepsis	female	42	32	1	0.5	0.56 (1.01)	**0.15 (0.34)**
13,703	bacteraemia	male	54	256	2	1	**0.16 (0.27)**	**0.16 (0.37)**
7914	OI	male	54	256	1	0.25	**0.25 (0.38)**	**0.26 (0.37)**
905	endocarditis	male	75	512	2	1	**0.31 (0.52)**	**0.27 (0.44)**
6037	urosepsis	male	86	1024 ^c^	1	1	**0.27 (0.51)**	**0.12 (0.25)**
7183	urosepsis	male	76	1024	2	1	**0.31 (0.55)**	**0.38 (0.52)**
3043	OI	female	74	>1024	2	0.5	N. A.	N. A.
8653	OI	male	56	>1024	2	1	N. A.	N. A.
3062	urosepsis	male	77	>1024	4	1	N. A.	N. A.
9190	wound infection	female	59	>1024	4	1	N. A.	N. A.

^a^ *E. faecalis* isolates were obtained from the Institute of Medical Microbiology in Jena, Germany. All clinical isolates originated from blood cultures except for 67230 and 245, which were sampled by swabs from infected mitral valves. / = N.A. to the laboratory standard strain ATCC 29,212. ^b^ OI = opportunistic infection. ^c^ In some isolates, ceftriaxone treatment led to trailing MIC endpoints, with wells showing the same level of turbidity observed in the growth control; followed by wells with less, but still visible, turbidity; and eventually wells with pinpoint growth (small aggregates). According to the EUCAST reading guide for broth microdilution [16], pinpoint growth was disregarded and recorded as the MIC. ^d^ FICI values are given as the lowest FICI with the median FICI in parentheses. Synergistic FICI values (lowest FICI ≤ 0.5 and median FICI ≤ 0.8) are indicated in bold. N.A. = not determined due to MIC > 1024 mg/L.

**Table 2 microorganisms-09-02150-t002:** Lowest sub-MIC CRO concentrations resulting in two-, four-, and eight-fold reductions in the effective MIC_PEN_ or MIC_AMP_ and corresponding FICI values of the resulting concentration combinations.

Isolate	MIC_CRO_ Alone [mg/L]	Lowest CRO Concentration Resulting in Two-Fold Reduction in MIC_PEN_ or MIC_AMP_				Lowest CRO Concentration Resulting in Four-Fold Reduction in MIC_PEN_ or MIC_AMP_				Lowest CRO Concentration Resulting in Eight-Fold Reduction in MIC_PEN_ or MIC_AMP_			
		Penicillin		Ampicillin		Penicillin		Ampicillin		Penicillin		Ampicillin	
		CRO conc. [mg/L]	FICI ^a^	CRO conc. [mg/L]	FICI	CRO conc. [mg/L]	FICI	CRO conc. [mg/L]	FICI	CRO conc. [mg/L]	FICI	CRO conc. [mg/L]	FICI
5187	1–2	1	1.50	0.5	1.00	1	1.25	0.5	0.75	N.A.	N.A.	0.5	0.625
245	2	0.5	0.75	0.5	0.75	1	0.75	0.5	**0.50**	N.A.	N.A.	0.5	**0.375**
26,786	2	0.25	0.625	0.5	0.75	1	0.75	1	0.74	1	0.625	N.A.	N.A.
404	4	1	0.75	1	0.75	2	0.75	1	**0.49**	N.A.	N.A.	2	0.625
ATCC	4	1	0.75	0.5	0.625	2	0.75	1	**0.50**	N.A.	N.A.	2	0.625
6747	4	1	0.75	0.5	0.625	2	0.75	1	**0.50**	N.A.	N.A.	2	0.625
11,223	8	1	0.625	1	0.625	1	**0.375**	1	**0.375**	4	0.625	4	0.625
22,424	8	1	0.625	1	0.625	2	**0.5**	1	**0.375**	4	0.625	2	**0.375**
8669	8	1	0.625	0.5	0.563	4	0.75	1	**0.375**	N.A.	N.A.	2	**0.375**
9367	8	1	0.625	0.5	0.563	2	**0.50**	1–2	**0.50**	4	0.625	2	**0.375**
1653	8	1	0.625	0.5	0.563	1–2	**0.50**	1	**0.375**	4	0.625	2–4	0.625
848	8	1	0.625	0.5–1	0.625	1	**0.375**	1	**0.375**	4	0.625	2	**0.375**
5597	8–16	1	0.625	0.5	0.563	2	**0.50**	2	**0.25**	8	1.125	4	0.625
10,485	16	1	0.56	0.25	0.52	2	**0.375**	1	**0.31**	4	**0.375**	2	**0.25**
6886	16	1	0.563	0.5–1	0.563	4	**0.50**	1	**0.313**	4	**0.375**	4	**0.375**
281	16	1–2	0.625	0.5	0.53	4–8	0.75	1	**0.313**	8	0.625	4	**0.375**
67,230	16	1	0.563	0.5–1	0.563	2	**0.375**	1–2	**0.375**	4	**0.375**	4	**0.375**
2164	16	1	0.563	1	0.563	2	**0.375**	2–4	**0.50**	4	**0.375**	8	0.625
4497	32	1	0.53	0.5	0.563	2–4	**0.375**	2	**0.313**	8	0.375	4	**0.25**
10,021	32	2	0.56	0.125	**0.50**	N.A.	N.A.	0.5	**0.266**	N.A.	N.A.	2	**0.18**
13,703	256	1	**0.50**	1	**0.50**	1	**0.26**	1	**0.25**	16	0.25	8	0.16
7914	256	1	**0.50**	1	**0.50**	8	0.28	4	**0.26**	64	0.375	64	0.375
905	512	2–4	0.51	1	**0.50**	32	0.313	4–8	**0.266**	N.A.	N.A.	64	0.25
6037	1024	8	0.51	0.5	**0.50**	128	0.375	0.5–1	**0.251**	512	0.625	2–4	**0.13**
7183	1024	16	0.52	0.5–1	**0.50**	64	0.313	64	0.313	256	0.375	N.A.	N.A.
3043	>1024	512	N.A.	4	N.A.	1024	N.A.	128	N.A.	N.A.	N.A.	N.A.	N.A.
8653	>1024	512	N.A.	2	N.A.	N.A.	N.A.	N.A.	N.A.	N.A.	N.A.	N.A.	N.A.
3062	>1024	128	N.A.	32–64	N.A.	256–512	N.A.	256	N.A.	512–1024	N.A.	N.A.	N.A.
9190	>1024	64–128	N.A.	2	N.A.	512	N.A.	256	N.A.	N.A.	N.A.	N.A.	N.A.

^a^ Synergistic FICI values ≤ 0.5 obtained at free ceftriaxone trough concentrations (CRO conc. = 1 to 4 mg/L) are indicated in bold. Two-, four- and eight-fold reductions in MIC_PEN_ or MIC_AMP_ correspond to fractional inhibitory concentration (FIC) values (FIC_A_ = MIC_A_ combined/MIC_A_ alone; FIC_A_ + FIC_B_ = FICI_A/B_) of 0.5, 0.25 and 0.125. The respective FICs of ceftriaxone are calculated by dividing the lowest sub-MIC CRO concentration (CRO conc.) by the MIC_CRO_. N.A. = the respective reduction in MIC_PEN_ or MIC_AMP_ not reached.

## Data Availability

No further datasets are available.

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
