# Peer review of "In Vitro Synergism of Penicillin and Ceftriaxone against Enterococcus faecalis"

_microorganisms, 2021, doi:10.3390/microorganisms9102150_

Round 1

Reviewer 1 Report

Thieme et al. investigate the synergism of penicillin and ceftriaxone against Enterococcus faecalis using a checkerboard synergy analysis of penicillin/ceftriaxone and ampicillin/ceftriaxone in 20 clinical isolates of E. faecalis compared to a single laboratory strain to provide context for outpatient parenteral antibiotic therapy treatments for infective endocarditis. The study is brief but performed with proper conditions and controls. The statistical analyses are appropriate for the data being analyzed. The manuscript is well-written with no major edits required in language.

We have only minor comments and questions that should be address before acceptance:

  1. What was the justification for 20 clinical isolations that were chosen?
  2. How large of a faecalis strain collection of is available?
  3. Is it not possible to obtain strains with MICCRO between 32-1024 mg/mL. This is the most glaring omission in this manuscript. There needs to be justification for why these were not examined. Do they simply not exist?
  4. Please change the abstract to include 20 clinical isolates and 1 control strain, not 21 isolates as per line 16.

  1. The introduction brings up infective endocarditis, but only three of the clinical backgrounds are from cases with endocarditis. They should either include more endocarditis isolates or expand the breadth of their introduction.

  1. What is the basis for the interpretation of the FICIs on lines 86-89?

  1. On line 101 they describe a weak correlation (rs = 0.56), which is the correct terminology, but on lines 104-105, they describe the correlation as “significant”, which is true based on P values, but not how Spearman’s rank correlation is typically described. It should something like, “strong inverse correlation”.

Reviewer 2 Report

Thieme et al. investigated whether benzylpenicillin/ceftriaxone treatment for E. faecalis is suitable for outpatient parenteral antibiotic therapy as ampicillin, which is usually used in combination to ceftriaxone, is not stabil during 24-hour continuous infusion. The manuscript is well written and easy to understand. Methods applied seem to be adequate. Nevertheless, one major concern is the reason for conducting this study. Ampicillin, which needs to be applied i.v. due to bioavailability reasons, could be replaced by amoxicillin via oral application to ensure consistently high drug levels. The amoxicillin oral/cefriaxone i.v. combination would serve as a suitable outpatient therapy.  It is not clear to the reader, why benzylpenicillin comes into play here. This needs to be addressed more precisely.

Minor points:

-line 95: E. faecalis-> write in italics

-please provide an additional table including antimicrobial susceptibility testing results of all included E. faecalis , which should be available as all isolates were obtained from blood culture samples

Round 2

Reviewer 2 Report

The authors have at least provided Table S1, but did not change the manuscript regarding the major concern I raised. From a clinical point of view there is no need to conduct this study, although it is performed using adequate methods. Therefore I would recommend to include a clear statement, why this (clinically not reasonable) study was performed. 

Author Response

Dear Reviewer,

thanks for your fast reply. However I am not sure whether you saw the attached file with our comments. Please find the comment where we adressed your main concern again. Do you think we should incorporate it in the manuscript's introduction?

Best regards,

Lara Thieme

The use of oral antibiotics as part of the treatment of endocarditis has only been assessed in one trial that we are aware of (Iversen et al. Partial Oral versus Intravenous Antibiotic Treatment of Endocarditis. NEJM 2019;380:415-424). This trial enrolled a tightly defined group of patients with left sided endocarditis (only 20% of those who were screened were enrolled). Patients with E. faecalis endocarditis were enrolled however there is limited detail in the paper as to the underlying details of these patients so the generalisability to the wider group of patients with E. faecalis endocarditis is uncertain. Patients in this trial had plasma levels of the orally administered antibiotics assessed on day 1 and 5 of the oral antibiotic treatment, with antibiotic doses adjusted according to pharmacokinetic finings. Patients with E. faecalis endocarditis were treated with a variety of oral antibiotic combinations, the most common of which were amoxicillin + moxifloxacin, amoxicillin + linezolid and amoxicillin + rifampicin. Oral amoxicillin + IV ceftriaxone has not been tested in a trial situation as far as we are aware. With the above, we feel that the use of oral antibiotic treatment for patients with E. faecalis endocarditis should only be undertaken as part of a very carefully designed trial.         
Intravenous benzylpenicillin is used as an alternative to intravenous ampicillin/amoxicillin for the treatment of E. faecalis endocarditis. The use of intravenous benzylpenicillin + ceftriaxone for the outpatient parenteral antibiotic therapy guided treatment of E. faecalis endocarditis is common in New Zealand and is also used in Australia and America. The use of this treatment combination has been reported previously (Tritle et al. Clin Infect Dis 2020;70:1263-4, Suzuki et al. Infect Dis 2020;52:135-8, Briggs et al. J Antimicrob Chemother 2021;76:2168-2171, Ingram et al. JAC Antimicrob Resist 2021;3:dlab128).

Round 3

Reviewer 2 Report

Thank you for your answer. I noticed your reply and I think that it is a good idea to incorporate a short paragraph in the introduction section, since the clinical relevance is not clear up to now.

Author Response

Dear reviewer,

we have now included more background information in the introduction. Please find the revised manuscript attached.

Best regards,

Lara Thieme